# Involvement of PI3K Pathway in Glioma Cell Resistance to Temozolomide Treatment

**DOI:** 10.3390/ijms22105155

**Published:** 2021-05-13

**Authors:** Adrian Zając, Joanna Sumorek-Wiadro, Ewa Langner, Iwona Wertel, Aleksandra Maciejczyk, Bożena Pawlikowska-Pawlęga, Jarosław Pawelec, Magdalena Wasiak, Monika Hułas-Stasiak, Dorota Bądziul, Wojciech Rzeski, Michał Reichert, Joanna Jakubowicz-Gil

**Affiliations:** 1Department of Functional Anatomy and Cytobiology, Institute of Biological Sciences, Maria Curie-Sklodowska University, Akademicka 19, 20-033 Lublin, Poland; adrian.zajac@poczta.umcs.lublin.pl (A.Z.); jsumorek@poczta.umcs.lublin.pl (J.S.-W.); olamaciejczyk77@gmail.com (A.M.); bozena.pawlikowska-pawlega@poczta.umcs.lublin.pl (B.P.-P.); monika.hulas-stasiak@poczta.umcs.lublin.pl (M.H.-S.); rzeskiw@hektor.umcs.lublin.pl (W.R.); 2Department of Medical Biology, Institute of Rural Health, Jaczewskiego 2, 20-950 Lublin, Poland; ewa.langner@gmail.com; 3Independent Laboratory of Cancer Diagnostics and Immunology, 1st Chair and Department of Oncological Gynaecology and Gynaecology, Medical University of Lublin, Staszica 16, 20-081 Lublin, Poland; iwonawertel@umlub.pl; 4Institute Microscopy Laboratory, Maria Curie-Sklodowska University, Akademicka 19, 20-033 Lublin, Poland; jaroslaw.pawelec@poczta.umcs.lublin.pl; 5Department of Pathological Anatomy, National Veterinary Research Institute, 57 Partyzantow Avenue, 24-100 Pulawy, Poland; magdalena.wasiak@piwet.pulawy.pl (M.W.); reichert@piwet.pulawy.pl (M.R.); 6Department of Biology, Institute of Medical Sciences, Medical College of Rzeszow University, Rejtana 16 C, 35-959 Rzeszów, Poland; dbadziul@ur.edu.pl

**Keywords:** gliomas, LY294002, temozolomide, programmed death, ER stress, Hsp27

## Abstract

The aim of the study was to investigate the anticancer potential of LY294002 (PI3K inhibitor) and temozolomide using glioblastoma multiforme (T98G) and anaplastic astrocytoma (MOGGCCM) cells. Apoptosis, autophagy, necrosis, and granules in the cytoplasm were identified microscopically (fluorescence and electron microscopes). The mitochondrial membrane potential was studied by flow cytometry. The activity of caspases 3, 8, and 9 and Akt was evaluated fluorometrically, while the expression of Beclin 1, PI3K, Akt, mTOR, caspase 12, and Hsp27 was determined by immunoblotting. SiRNA was used to block Hsp27 and PI3K expression. Cell migration and localization of Hsp27 were tested with the wound healing assay and immunocytochemistry, respectively. LY294002 effectively diminished the migratory potential and increased programmed death of T98G and MOGGCCM. Autophagy was dominant in MOGGCCM, while apoptosis was dominant in T98G. LY294002 with temozolomide did not potentiate cell death but redirected autophagy toward apoptosis, which was correlated with ER stress. A similar effect was observed after blocking PI3K expression with siRNA. Transfection with Hsp27 siRNA significantly increased apoptosis related to ER stress. Our results indicate that inhibition of the PI3K/Akt/mTOR pathway sensitizes glioma cells to apoptosis upon temozolomide treatment, which was correlated with ER stress. Hsp27 increases the resistance of glioma cells to cell death upon temozolomide treatment.

## 1. Introduction

Gliomas are the most common and most malignant primary tumors of the central nervous system. They are characterized by aggressive proliferation and growth as well as diffusive infiltration into adjacent brain parenchyma and resistance to cell death [1,2,3,4]. For these reasons, the outcome of patients with malignant gliomas is poor and, unfortunately, there are no effective therapeutic treatments. Surgery followed by radiotherapy in combination with up to six maintenance cycles of temozolomide chemotherapy constitute the standard of care for the majority of patients with newly diagnosed glioblastoma [5,6]. Hence, it is crucial to explore and develop novel and more effective therapeutic agents that can prevent the growth of glioblastoma cells.

A vast majority of malignant gliomas possess alterations in the PI3K/Akt/mTOR pathway, and its activation has been clearly validated as an essential step for the initiation and maintenance of the tumorigenic phenotype. In consequence of the upregulation of this pathway, a cascade of biological events is activated from cell growth and proliferation to survival and migration, which drive tumor progression, survival, angiogenesis, and metastasis. This pathway also plays a major role in the resistance of tumor cells to conventional therapy [7,8,9,10]. One of the PI3K inhibitors is LY294002, a classic ATP-competitive agent which acts by binding to the ATP-binding cleft of the lipid kinase [11,12]. It is also known as an autophagy blocker. Therefore, it is a powerful tool to study the role of PI3K in human tumorigenesis and evaluate the potential utility of enzyme modulators as potential anticancer therapeutics [11,12,13,14,15]. However, a large number of clinical trials in which targeted therapy was tested have indicated that monotherapy is not sufficient and has limited efficacy. Therefore, combination therapy has greater potential [2,16,17].

Temozolomide is an oral alkylating drug that penetrates the brain and does not require hepatic metabolism for activation. Its molecular action is based on the formation of O6-methylguanine, which in consequence mispairs with thymine, leading to cell cycle arrest at G2/M and consequent programmed cell death. However, glioma cells develop a drug resistance mechanism that decreases the efficacy ratio [1,18].

Therefore, the aim of the present study was to investigate the effect of LY294002 on the efficacy of temozolomide treatment in human glioma cells. Special attention was paid to cell death induction and the limitation of the migratory potential. The molecular mechanism of cancer resistance to such treatment was also evaluated.

## 2. Results

### 2.1. Cell Death and Migratory Potential after LY294002 and Temozolomide Treatment

The MOGGCCM and T98G cells were incubated with different LY294002 concentrations (5–30 µM) for 6, 12, 24, and 48 h and stained with dyes typical for apoptosis, autophagy, and necrosis, namely Hoechst 33342, acridine orange (AO), and propidium iodide, respectively. As shown in Figure 1, the highest number of apoptotic and autophagic cells without a significant increase in the number of necrotic ones was observed after the 24 h incubation with a concentration of 10 µM (Figure 1e,f). Therefore, this concentration was chosen for further studies with simultaneous application with temozolomide (100 µM for MOGGCCM or 50 µM for T98G, as indicated in earlier experiments [19,20]).

As presented in Figure 2, the simultaneous application of both drugs increased the number of apoptotic cells in comparison to the separate treatment. Apoptosis in the MOGGCCM cells reached 8% after the incubation with LY294002, 4% after the treatment with temozolomide alone, and ca. 25% after administration of both drugs. In T98G cells, apoptosis was initiated in 28% of the cells by LY294002 and in 11% of the cells by temozolomide. Both drugs induced apoptosis in 31% of the cells, which was only a slightly higher value in comparison to LY294002 alone. Autophagy in MOGGCCM was dominant and observed in ca. 40% of the cells after administration of LY294002 alone and in combination with temozolomide treatment, while temozolomide alone initiated this type of death in 11% of the cell population. In the T98G line, autophagy was observed only after the LY294002 treatment (19%). Necrosis was not observed in either of the studied cell lines.

As it was revealed by wound assay, temozolomide alone and in combination with LY294002 significantly decreased the migratory potential of the MOGGCCM and T98G cells (Figure 3). In both cell lines, simultaneous application of drugs was the most effective, reducing the number of cells within the wound to 42% in anaplastic astrocytoma and 36% in glioblastoma cells. The antimigratory potential of LY294002 was cell-line-specific and appeared to be more effective in T98G. Temozolomide showed the weakest, but still very significant, properties in limiting cell mobility, decreasing the number of cells within the wound to about 50%.

### 2.2. The Effect of Caspase Activity, Mitochondrial Membrane Potential, and Beclin 1 Expression

Apoptosis, autophagy, and necrosis are characterized by changes at the subcellular and molecular levels. Therefore, to confirm the reliability of the results obtained from microscopic observations, we decided to evaluate the effect of LY294002 and temozolomide on the activity of proapoptotic caspases using the colorimetric method, the level of pro-autophagic Beclin 1 by immunoblotting, and the value of mitochondrial membrane potential by flow cytometry. The activity of caspases upon the LY294002 and temozolomide treatment was cell-type-specific (Figure 4a,b). In the MOGGCCM cell line, activation of caspases was observed only after the separate temozolomide incubation (caspase 3) and in combination with LY294002 (caspases 3, 8, and 9) (Figure 4a). The treatment with both drugs was correlated with a significant decrease in the mitochondrial membrane potential (Figure 5a,c). The T98G cells were more sensitive to the LY294002 and temozolomide treatment (both alone and in combination), and a significant increase in activity of caspases 3 and 9 was observed in each experimental variant (Figure 4b). The activation of caspases was accompanied by a decline in the mitochondrial membrane potential (Figure 5b,d). No caspase 8 activation was observed. In the case of the autophagy marker Beclin 1, an elevated level of the protein was observed in the MOGGCCM cells after the separate treatment with LY294002 or temozolomide and in T98G only after the LY294002 incubation. The treatment with both drugs decreased the level of Beclin 1 in the MOGGCCM cells with no significant effect in the T98G cells.

### 2.3. Effect of LY294002 and Temozolomide on the PI3K/Akt/mTOR Pathway

To study the effect of LY294002 and temozolomide on the prosurvival PI3K/Akt/mTOR pathway, immunoblotting analysis was used to present the level of proteins, while the activity of Akt was studied colorimetrically. As shown in Figure 6a, LY294002 alone and in combination with temozolomide significantly inhibited the expression of PI3K in MOGGCCM (by ca. 37%) and T98G (by ca. 18%). In contrast, temozolomide alone did not exhibit such inhibitory potential, and the level of the protein was comparable with that in the control. A similar correlation was observed in the case of Akt expression (Figure 6b). LY294002 alone and in combination with temozolomide inhibited the expression of the kinase in both cell lines. Temozolomide alone did not change the level of the protein but, surprisingly, significantly inhibited the activity of Akt. This was also observed after LY294002 treatment, both alone and in combination with temozolomide (Figure 6d). LY294002 exhibited the highest effectiveness in the MOGGCCM cells (reduction of 35%), whereas the combined LY294002 and temozolomide treatment was most effective in the T98G cells (reduction of 58%). LY294002 and temozolomide also appeared to be potent inhibitors of the mTOR protein expression (Figure 6c). LY294002 was the most effective inhibitor, reducing the level of the protein by 85% in the MOGGCCM cells and 90% in the T98G cells. Temozolomide alone in MOGGCCM was the least effective, reducing the level of mTOR by 8%. Our results also indicated that both drugs significantly reduced the Akt activity.

### 2.4. Blocking the Expression of PI3K

To obtain direct proof of the involvement of PI3K in glioma cell resistance, silencing with specific anti-PI3K siRNA was used in the experiments (Figure 7). The reduced level (Figure 7a) of the kinase and the diminished activity of downstream Akt (Figure 7b) were correlated with very significant apoptosis induction in the transfected cells after the temozolomide treatment (Figure 7c). Approximately 70% of MOGGCCM and 30% of T98G cells exhibited symptoms of programmed death. Interestingly, neither autophagy nor necrosis was observed.

### 2.5. Effect of LY294002 and Temozolomide on Granule Formation

Staining of the MOGGCCM and T98G cells treated with LY294002 and temozolomide with acridine orange revealed the presence of round-shaped granules within the cytoplasm, which were not stained with AO (AO-negative) (Figure 8a,b). In MOGGCCM, the process was most frequent after the combined temozolomide and LY294002 treatment (ca. 33% of cells), while temozolomide alone initiated granule formation in only 2%. No such structures were observed after the LY294002 separate application. The T98G cells were more resistant than MOGGCCM, and the number of cells with vesicular structures did not exceed 8% (temozolomide treatment). The results obtained after the AO staining were confirmed using the electron microscopy technique, which showed large vesicles without any cellular structural elements inside (Figure 8c). As revealed by immunoblotting technique, the presence of vesicles was correlated with an elevated level of caspase 12, a well-known marker of endoplasmic reticulum (ER) stress (Figure 8d).

The immunocytochemistry method showed that the vesicles were surrounded by Hsp27 (Figure 9a,b). The structure of the endoplasmic reticulum was more granular in the vicinity of the vesicular structures (Figure 9c,d).

### 2.6. Involvement of Hsp27 Expression in Glioma Cell Resistance after the LY294002 and Temozolomide Treatment

Hsp27 belongs to the family of molecular chaperones, the overexpression of which in cancer cells is responsible for chemoresistance and poor prognosis for patients in consequence [19,20]. Immunoblotting technique showed that LY294002 applied alone and in combination with temozolomide inhibited the level of Hsp27 expression in the MOGGCCM cells (Figure 10a). No correlation was observed after the temozolomide treatment. In the case of T98G, a decreased level of Hsp27 was observed after the incubation with both compounds. The inhibition of the expression of this molecular chaperone with specific siRNA resulted in a microscopically observed significant increase in the level of apoptotic cells in all experimental variants (Figure 10b,d). The most effective treatment was the simultaneous application of LY294002 and temozolomide in the transfected MOGGCCM and T98G cells, and the percentages of dead cells were 52% and 62%, respectively. It was also evident that the T98G cells with the blocked expression of Hsp27 were more sensitive to apoptosis induction after LY294002 or temozolomide than the MOGGCCM cells. The inhibition of Hsp27 expression was also correlated with a significant increase in the number of cells with granules. Once again, LY294002 and temozolomide applied in combination was the most effective variant (Figure 10c). This was correlated with increased expression of caspase 12 (Figure 10e) and decreased activity of Akt (Figure 10f).

## 3. Discussion

The phosphatidylinositol 3-kinase (PI3K kinase) family of genes encodes kinases regulating multiple cellular processes that are frequently deregulated in cancer. These include cell survival, proliferation, chemoresistance, cell cycle progression, angiogenesis, invasion, and metastasis. Deregulation of the PI3 kinase signaling pathway has been observed frequently in human glioma biopsy and cell lines. Many studies have identified that the PI3K/Akt/mTOR axis is overactive in GBM, which increases tumor resistance to treatment [8,21]. The current therapies yield only poor responses and survival rates with median survival for high-grade glioma patients of approximately 15 months [4,22,23]. This highlights the current need for novel treatments. Therefore, PI3 kinase inhibitors are now emerging with the potential for the treatment of cancers [15,24,25,26]. There are few reports on the use of LY294002 in investigations of gliomas. This inhibitor was found to sensitize cultured primary glioblastoma cells obtained from surgical specimens to chemotherapy-induced cell death. It also inhibited glioma cell growth and proliferation, which was correlated with an increased number of apoptotic cells and caspase 3 activation [16,27]. In our experiments, 10 µM of LY294002 applied for 24 h appeared to eliminate T98G and MOGGCCM cells via programmed death (but not necrosis) very effectively. Autophagy was dominant in the astrocytoma cells. At the molecular level, it was correlated with Beclin 1 overexpression. The process was accompanied by reduced mitochondrial membrane potential. It was observed that damaged mitochondria lost their transmembrane potential, which initiated the type of autophagy called mitophagy, i.e., selective removal of damaged mitochondria by autophagosomes and their subsequent catabolism by lysosomes [28]. Autophagy serves a housekeeping function, maintaining cellular homeostasis and quality control of cellular components by facilitating the clearance or turnover of long-lived or misfolded proteins, protein aggregates, and damaged organelles [29,30]. If autophagy is inhibited, accumulation of damaged intracellular organelles is observed; protein aggregation and deficiency of the correct energy supply bring about cell death. However, cancer cells from multiple tumor types have been shown to have high basal autophagy, which may be used as a cell survival mechanism. Therefore, in this context, autophagy is not a desirable type of death [31,32]. In our experiments, autophagy was not observed in the T98G cells treated with LY294002, whereas apoptosis dominated. This was correlated with decreased MMP, which served as an apoptotic marker, and confirmed by the increased activity of caspases 3 and 9.

Many studies have revealed that the inhibition of autophagy enhances apoptosis induced by the blockers of the PI3K/Akt/mTOR pathway [32,33,34]. In our experiments, LY294002 decreased the level of PI3K, Akt, and mTOR expression as well as Akt activity in both cell lines, which was correlated with apoptosis and autophagy induction. This may indicate the involvement of this pathway in both cell death processes [35]. A different effect on the level of PI3K/Akt/mTOR expression was observed after the temozolomide treatment. It had no impact on the level of PI3K and Akt, but surprisingly, it inhibited Akt activity and the expression of downstream mTOR in the MOGGCCM and T98G cells. This was correlated with autophagy induction in the MOGGCCM cell line and apoptosis in T98G, but to a smaller extent than after the LY294002 treatment. Both types of cell death were accompanied by the activation of specific markers: Beclin 1 in the case of autophagy and caspases 3 and 9 in the case of apoptosis. The decreased sensitivity of the studied cells to apoptosis or autophagy may be correlated with the absence of inhibition of PI3K and Akt expression, which is in agreement with results reported by other authors, who observed even an increased level of Akt after temozolomide treatment and, in consequence, chemoresistance [36]. There are also suggestions that simultaneous inhibition of the expression of PI3K or Akt with mTOR is more effective in the elimination of glioma cells via programmed death than a single kinase of the pathway [37].

Some articles suggest that PI3K/Akt/mTOR signal transmission may influence temozolomide-mediated cytotoxicity and chemoresistance [36,38]. In our experiments, the combined treatment of temozolomide with LY294002 had no significant effect on increasing the sensitivity of the T98G cell line to apoptosis induction, in comparison to the administration of LY294002 alone. A similar effect was observed in the MOGGCCM cells, but there was a difference in the proportion of apoptotic and autophagic cells. Both drugs decreased the number of autophagic cells in favor of the apoptotic ones, but autophagy was still dominant. A similar tendency in apoptosis and autophagy initiation in MOGGCCM and T98G cells was observed after combined temozolomide and quercetin treatment [19,20], but it seems that LY294002, a synthetic quercetin derivative, is more efficient. To obtain direct proof of the involvement of PI3K in MOGGCCM and T98G sensitivity to cell death upon the temozolomide treatment, we decided to block the expression of the kinase with specific siRNA. The results showed that the LY294002 replacement by siRNA did not increase the sensitivity of the cell lines to cell death initiation after the temozolomide incubation, but redirected the death signal into apoptosis rather than autophagy.

The microscopic observation of the MOGGCCM and T98G cells revealed small round granules in the cytoplasm and ER after the temozolomide treatment, both alone and in combination with LY294002. They were visibly surrounded by Hsp27. This was accompanied by increased caspase 12 expression. All the data suggest that this might be correlated with the ER stress response described by us in previous studies on quercetin and temozolomide [19,20]. Hsp27 is one of the most widely studied heat shock proteins in tumor diseases, and its enhanced expression inhibits ER stress and apoptosis and leads to cell survival [39]. These molecular chaperones act as negative prognostic markers. Their expression rises with the grade of the tumor and may participate in oncogenesis. Therefore, inhibition of Hsp expression has become a strategy for cancer therapy [40]. It has been demonstrated that Akt and Hsp27 exist in a signaling complex with direct protein–protein interaction. Removal of Hsp27 from the Akt signal module prevented kinase activity and resulted in accelerated apoptosis [41,42]. In our experiments, blocking the Hsp27 expression in MOGGCCM and T98G cells by specific siRNA resulted in decreased Akt activity and increased apoptosis induction upon the LY294002 and temozolomide treatment. In the case of MOGGCCM, the level of this type of death was comparable with that in the nontransfected cells and those with blocked PI3K expression. No significant autophagy was observed. In T98G, the proapoptotic effect after blocking the Hsp27 expression was more spectacular than in cells transfected with PI3K siRNA. Taking into consideration that the overexpression of Hsp27 in high-grade glioma plays a very important protective and chemoresistant role, such results would be beneficial. The decreased level of Hsp27 also resulted in an increased percentage of cells with granules in the cytoplasm. This may suggest that elimination of this molecular chaperone increased the accumulation of misfolded protein in the cells, which may have led to ER stress, especially as there was additional overexpression of caspase 12. Another interesting aspect is the absence of granular structures in the MOGGCCM and T98G cells with blocked PI3K expression. This may be explained by the presence of the Hsp27 and Akt complex. Elimination of the kinase does not affect the level of Hsp27, which may perform its chaperone functions unhindered.

Our results indicated that the combination of temozolomide and the synthetic derivative of quercetin, i.e., LY294002, significantly inhibited glioma cell migration and exhibited high potential in the elimination of glioma cells via programmed death. The inhibition of the PI3K/Akt pathway led to apoptosis rather than autophagy in the studied cells. The cell death was preceded by ER stress. The presence of Hsp27 increased the resistance of glioma cells to cell death upon the temozolomide treatment; therefore, considering inhibition thereof during cancer treatment would be beneficial.

## 4. Materials and Methods

### 4.1. Cells and Culture Conditions

Human glioblastoma multiforme cells (T98G, European Collection of Cell Cultures, ECACC, Porton Down, Salisbury, UK) and human anaplastic astrocytoma cells (MOGGCCM, European Collection of Cell Cultures ECACC, Porton Down, Salisbury, UK) were grown in a 3:1 mixture of Dulbecco’s Modified Eagle Medium (DMEM) and Ham’s nutrient mixture F-12 (Sigma, St. Louis, MO, USA) supplemented with 10% fetal bovine serum (Sigma, St. Louis, MO, USA), penicillin (100 units/mL) (Sigma, St. Louis, MO, USA), and streptomycin (100 µg/mL) (Sigma, St. Louis, MO, USA). The cultures were kept at 37 °C in a humidified atmosphere of 95% air and 5% CO_2_.

### 4.2. Drug Treatment

Temozolomide (Schering-Plough) (100 µM for MOGGCCM or 50 µM for T98G) and LY294002 (Sigma, St. Louis, MO, USA) (5, 10, 20, and 30 µM) were dissolved in DMSO. The temozolomide doses and incubation times were chosen on the basis of earlier experiments [19,20]. In the case of LY294002, MOGGCCM and T98G cells were incubated with different inhibitor concentrations for 6, 12, 24, and 48 h. The concentrations used in simultaneous applications were chosen experimentally (data not shown). As controls, T98G and MOGGCCM cells were incubated only with 0.01% DMSO.

### 4.3. Microscopic Detection of Apoptosis, Autophagy, and Necrosis with Fluorochromes

For identification of apoptosis and necrosis, a staining method with Hoechst 33342 (Sigma, St. Louis, MO, USA) and propidium iodide (Sigma, St. Louis, MO, USA) was chosen, as described previously [19,20]. In the case of autophagy, staining with acridine orange (AO) to detect typical acidic vesicular organelles (AVOs) was performed. The same method facilitated observation of granules (“wholes”) in the cells [20]. A fluorescence microscope (Nikon E-800, Tokyo, Japan; excitation filter UV-2A Ex 330–380 for Hoechst 33342/propidium iodide and B 2A Ex 450–490 for AVOs, objective 40×/0.75, camera Nikon D-200, Tokyo, Japan) was used for morphological analysis of dead cells. Cells exhibiting blue fluorescent nuclei (fragmented or/and with condensed chromatin) after Hoechst 33342 staining were interpreted as apoptotic (Figure 2c). Cells exhibiting pink fluorescent nuclei upon propidium iodide were interpreted as necrotic. Typical AO-positive cells exhibiting granular discretion of AVOs in the cytoplasm were interpreted as autophagic (Figure 2d). Additionally, staining with AO facilitated distinguishing round-shaped granules within cytoplasm, not stained with fluorochrome (AO-negative, excitation filter G 2A Ex 510–560) (Figure 8b). At least 1000 cells in randomly selected microscopic fields were counted under the microscope. Each experiment was conducted in triplicate.

### 4.4. Detection of Mitochondrial Membrane Potential by Flow Cytometry

For the mitochondrial membrane potential (∆ψm) (MMP) analysis, staining with a 3,3′-dihexyloxacarbocyanine iodide (DiOC_6_(3)) fluorochrome was chosen according to the method described previously [19]. The control and drug-treated cells were incubated with the fluorochrome at the final concentration of 50 nM for 20 min at 37 °C in the dark, washed three times with PBS, and analyzed with the FacsCanto instrument (Becton Dickinson, San Jose, CA, USA) with channel and λEx/Em = 482/504. Each experiment was performed in triplicate.

### 4.5. ER Staining

A staining method with fluorochrome 3,3′-dihexyloxacarbocyanine iodide was used for identification of ER [20]. The cells were incubated with 10 µM of DiOC_6_(3) for 10 min in the dark at 37 °C. Morphological analysis was performed under a fluorescent microscope Nikon E-800, excitation filter B 2A Ex 450–490.

### 4.6. Cell Migration Test

Tumor cell migration was assessed by means of the wound assay model [19]. The cells were plated at 2.5 × 10^5^ cells on 4 cm diameter culture dishes (Nunc, Thermo Scientific, Waltham, MA USA). The next day, the cell monolayer was scratched with a pipette tip (P300, Bionovo, Legnica, Poland), the medium and dislodged cells were aspirated, and the plates were rinsed twice with PBS. Next, fresh culture medium was applied and the number of cells that migrated into the wound area after 24 h was estimated in the control and drug-treated cultures. The plates were stained with the May–Grünwald–Giemsa method. Briefly, after medium was discarded, the cells were incubated with May–Grünwald dye for 2 min and then incubated in dye diluted with an equal volume of water for another 2 min. Thereafter, the dye was removed and Giemsa stain, previously diluted (1 vol. Giemsa/19 vol. water), was added for 20 min. The dishes were rinsed three times with distilled water and dried. The observation was performed using Nikon E-800 and micrographs were prepared. Cells that migrated to the wound area were counted on micrographs using the Image J program.

### 4.7. Transmission Electron Microscopy

Control and treated cells were scraped from the culture flasks and fixed in 4% glutaraldehyde in 100 mM cacodylate buffer for 2 h and in 1% osmium tetroxide for the next 2 h, all at 4 °C. The cells were dehydrated in series of alcohol and acetone and embedded in LR White resin. Ultrathin sections were cut with a diamond knife on a microtome RMC MT-XL (Tucson, AZ, USA), collected on copper grids coated with formvar, and contrasted with the use of uranyl acetate and Reynold’s liquid. At least 100 cells were examined from each experimental variant. The samples were observed under a Zeiss Libra 120 transmission electron microscope (Carl Zeiss SMT AG, Oberkochen, Germany).

### 4.8. Immunoblotting

Whole-cell extracts were prepared by lysing the cells in hot buffer containing 125 mM Tris-HCl pH 6.8, 4% SDS, 10% glycerol, and 100 mM dithiothreitol (DTT). The protein concentration was measured with the Bradford method [43].

In total, 80 µg of proteins were separated by 10% SDS-PAGE [44] and transferred onto an Immmobilon P membrane (Sigma, St. Louis, MO, USA). After blocking with 3% low-fat milk for 1 h, the membranes were incubated overnight with primary antibodies: mouse anti-Hsp27 (StressGen, San Diego, CA, USA, dilution 1:1000), anti-Akt1 (Santa Cruz Biotechnology, Dallas, TX, USA, 1:500), rabbit anti-Beclin 1 (Santa Cruz Biotechnology, Dallas, TX, USA, 1:500), anti-caspase 12 (Cell Signaling, Danvers, MA, USA, dilution 1:1000), anti-PI3K (Santa Cruz Biotechnology, Dallas, TX, USA, 1:500), and anti-mTOR (Santa Cruz Biotechnology, Dallas, TX, USA, 1:500). After three washes with PBS enriched with 0.05% Triton X-100 (Sigma, St. Louis, MO, USA), the membranes were incubated with secondary antibodies conjugated with alkaline phosphatase (AP) for 2 h. Proteins were detected with AP substrates: 5-bromo-4-chloro-3-indolylphosphate (BCIP) and nitro-blue tetrazolium (NBT) (Sigma, St. Louis, MO, USA) in N,N-dimethylformamide (DMF, Sigma, St. Louis, MO, USA). The results obtained were analyzed qualitatively on the basis of the band thickness, width, and color depth. The quantitative analysis of protein bands was performed using the Bio-Profil Bio-1D Windows Application V.99.03 program. The data were normalized relative to β-actin (Sigma, St. Louis, MO, USA, working dilution 1:2000). Three independent experiments were performed.

### 4.9. Akt and Caspase Activity Assay

The activity of caspases 3, 8, and 9 was analyzed with a SensoLyte AMC Caspase Substrate Sampler Kit (AnaSpec, Fremont, CA, USA), and Akt was analyzed with AKT/ERK Activation InstantOne ELISATM Kit (Invitrogen by Thermo Fisher Scientific, Waltham, MA, USA) according to the manufacturer’s protocol and measured colorimetrically with a 2030 Multilabel Reader VictorTMx4 (Perkin Elmer, Waltham, MA, USA) microplate reader.

### 4.10. Indirect Immunofluorescence

After the LY294002 and/or temozolomide treatment, the cells were washed three times with PBS and fixed in 3.7% paraformaldehyde (Sigma, St. Louis, MO, USA) in PBS for 10 min. After extensive washing with PBS, the cells were treated with 0.2% Triton X-100 (Sigma, St. Louis, MO, USA) for 7 min and then washed three times with PBS, all at room temperature. Subsequently, a blocking step of 30 min incubation in 5% low-fat milk at room temperature was included. Then, the cells were incubated with anti-Hsp27 mouse monoclonal antibody (SPA 800, StressGen, San Diego, CA, USA) diluted 1:100. The primary antibodies were detected with FITC fluorescein isothiocyanate)-conjugated secondary anti-mouse antibodies (Sigma, St. Louis, MO, USA). Protein localization in the cells was analyzed using the fluorescence microscope PASCAL5 (Carl Zeiss SMT AG, Oberkochen, Germany) with the excitation wavelength typical for fluorescein λ = 488 nm. Three independent experiments were performed. Over 100 cells were analyzed in each experimental variant.

### 4.11. T98G and MOGGCCM Transfection with siRNA

The cells at a density of 2 × 10^5^ were incubated for 24 h at 37 °C in a CO_2_ incubator to reach 60–80% confluence. After washing with a 3:1 DMEM/Ham’s F-12 mixture without serum and antibiotics, the medium was aspirated. The cells were overlaid with transfection probes containing 2 µL of specific anti-PI3K small interfering RNA (siRNA) (Santa Cruz Biotechnology, Dallas, TX, USA) or anti-Hsp27 siRNA (Santa Cruz Biotechnology, Dallas, TX, USA) and 2 µL of Transfection Reagent (Santa Cruz Biotechnology, Dallas, TX, USA). After 5 h of incubation at 37 °C in a CO_2_ incubator, the medium was supplemented with a medium containing 20% fetal bovine serum and 200 µg/mL of antibiotics. Incubation for an additional 18 h was performed. After changing the medium to the fresh normal growth one, the transfected cells were taken for further experiments.

### 4.12. Statistical Analysis

A one-way ANOVA test followed by Dunnett’s multiple comparison analysis was used for statistical evaluation. *p* < 0.01 of data presented as mean ± standard deviation (SD) was taken as the criterion of significance.

## Figures and Tables

**Figure 1 ijms-22-05155-f001:**
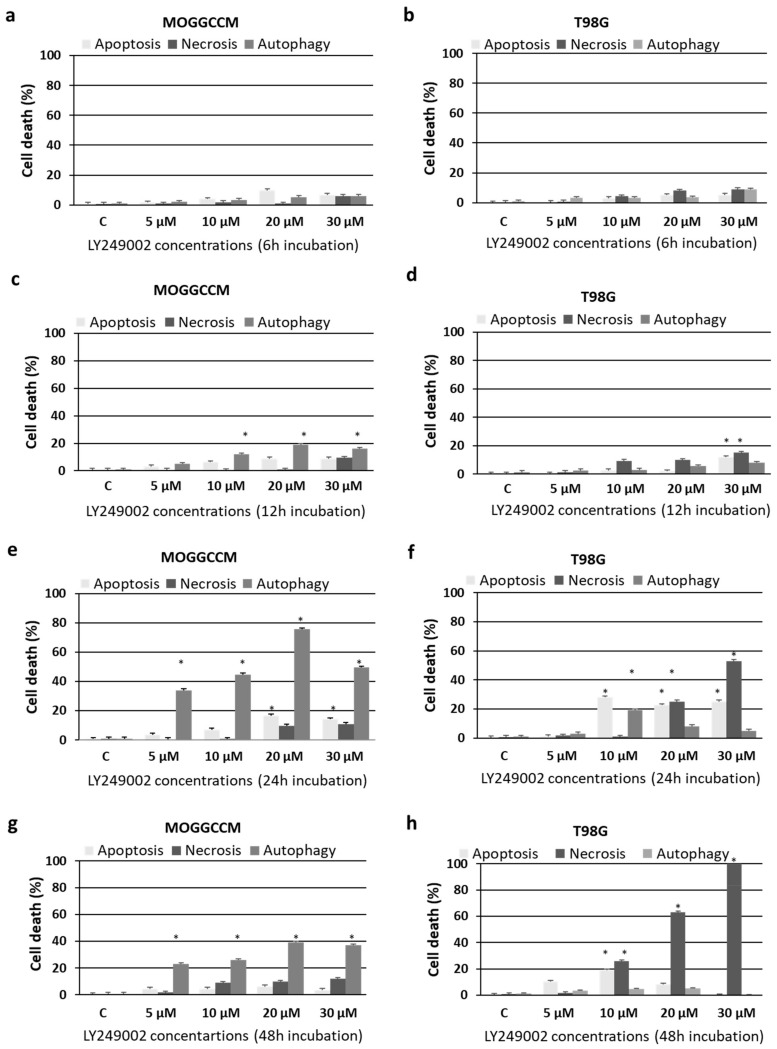
The effect of different concentrations of LY294002 on the levels of apoptosis, autophagy, and necrosis identified microscopically after staining with Hoechst 33342, acridine orange, and propidium iodide, respectively, in anaplastic astrocytoma (MOGGCCM; (**a**,**c**,**e**,**g**)) and glioblastoma multiforme cells (T98G; (**b**,**d**,**f**,**h**)) incubated with the compound for 6 (**a**,**b**), 12 (**c**,**d**), 24 (**e**,**f**), or 48 (**g**,**h**) h. C—control, * *p* < 0.01 compared to control.

**Figure 2 ijms-22-05155-f002:**
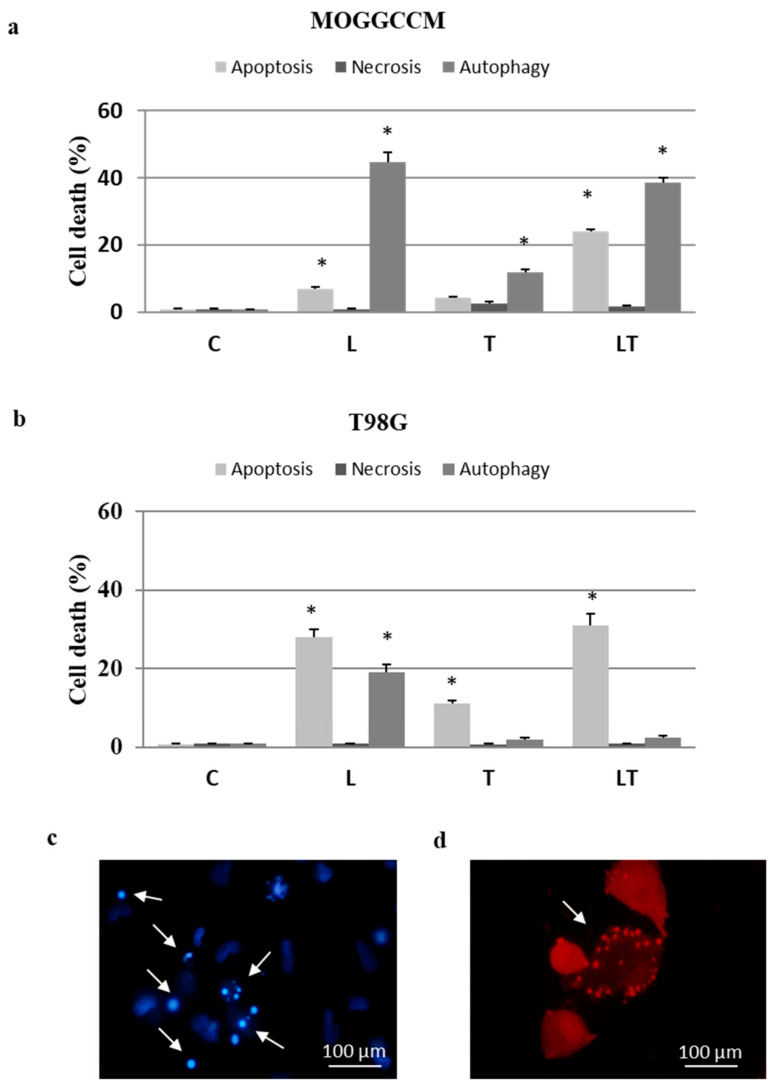
The effect of LY294002 (L) and temozolomide (T), applied alone and in combination (LT) on the levels of apoptosis, autophagy, and necrosis identified microscopically after staining with Hoechst 33342, acridine orange, and propidium iodide, respectively, in anaplastic astrocytoma (MOGGCCM; (**a**)) and glioblastoma multiforme (T98G; (**b**)) cells; (**c**) representative picture of apoptotic T98G cells (indicated by arrows) stained with Hoechst 33342 after LT treatment; (**d**) representative picture of autophagic MOGGCCM cells (indicated by arrow) stained with acridine orange after LT treatment. C—control, * *p* < 0.01 compared to control.

**Figure 3 ijms-22-05155-f003:**
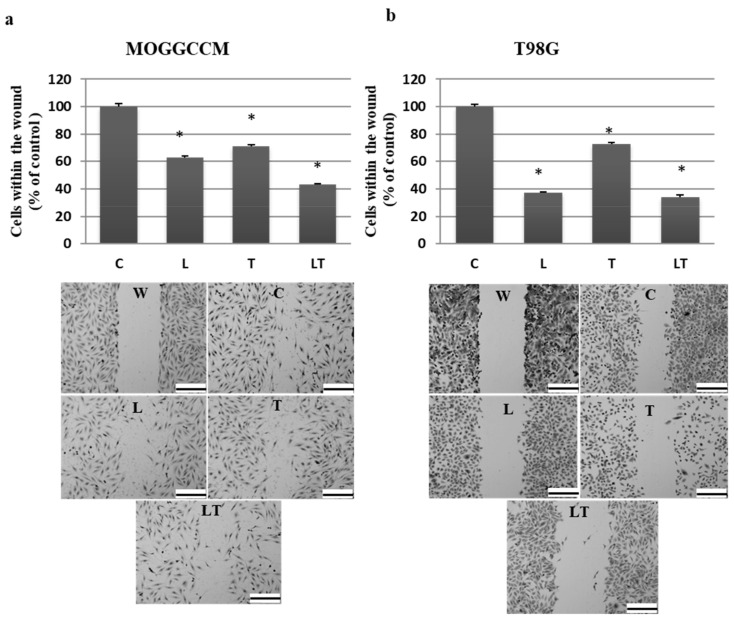
The effect of LY294002 (L) and temozolomide (T) applied alone and in combination (LT) on MOGGCCM (**a**) and T98G (**b**) cell migration estimated by wound assay. A quantitative analysis of migrated cells with representative pictures. W—wound, C—control, * *p* < 0.01 compared to control, scale bar—400 µm.

**Figure 4 ijms-22-05155-f004:**
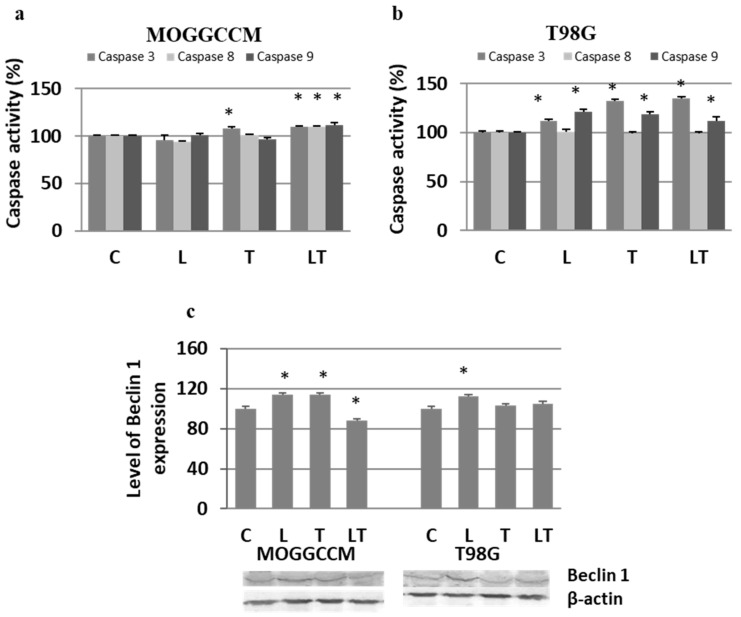
Activity of caspases 3, 8 and 9 after LY294002 (L) and temozolomide (T) treatment in MOGGCCM (**a**) and T98G (**b**) cells as well as Beclin 1 expression with representative blots (**c**). C—control, * *p* < 0.01 compared to control.

**Figure 5 ijms-22-05155-f005:**
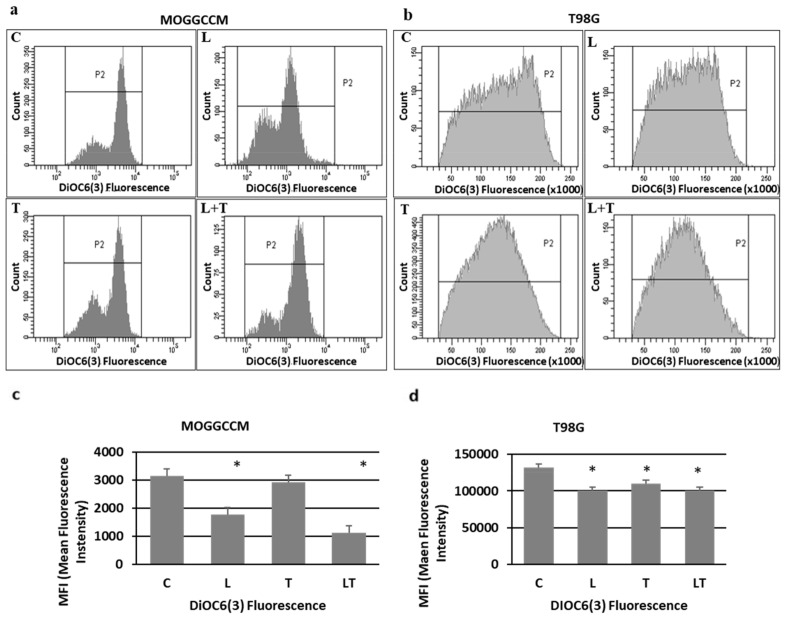
The effect of LY294002 (L) and temozolomide (T), applied alone or in combination (LT) on the mitochondrial membrane potential (MMP) expressed as DiOC_6_(3) mean fluorescence intensity as gates (histograms) (**a**,**b**) and diagrams (**c**,**d**) in MOGGCCM (**a**,**c**) and T98G (**b**,**d**) cells. C—control, * *p* < 0.01 compared to control.

**Figure 6 ijms-22-05155-f006:**
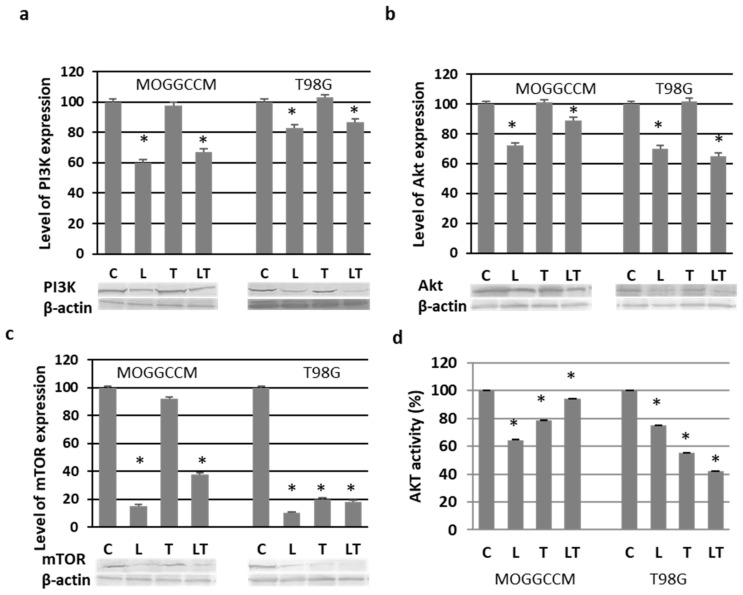
Levels of PI3K (**a**), Akt (**b**), and mTOR (**c**) with representative blots as well as Akt activity (**d**) in MOGGCCM and T98G cells after LY294002 (L) and temozolomide (T) incubation, applied alone or in combination (LT). C—control, * *p* < 0.01 compared to control.

**Figure 7 ijms-22-05155-f007:**
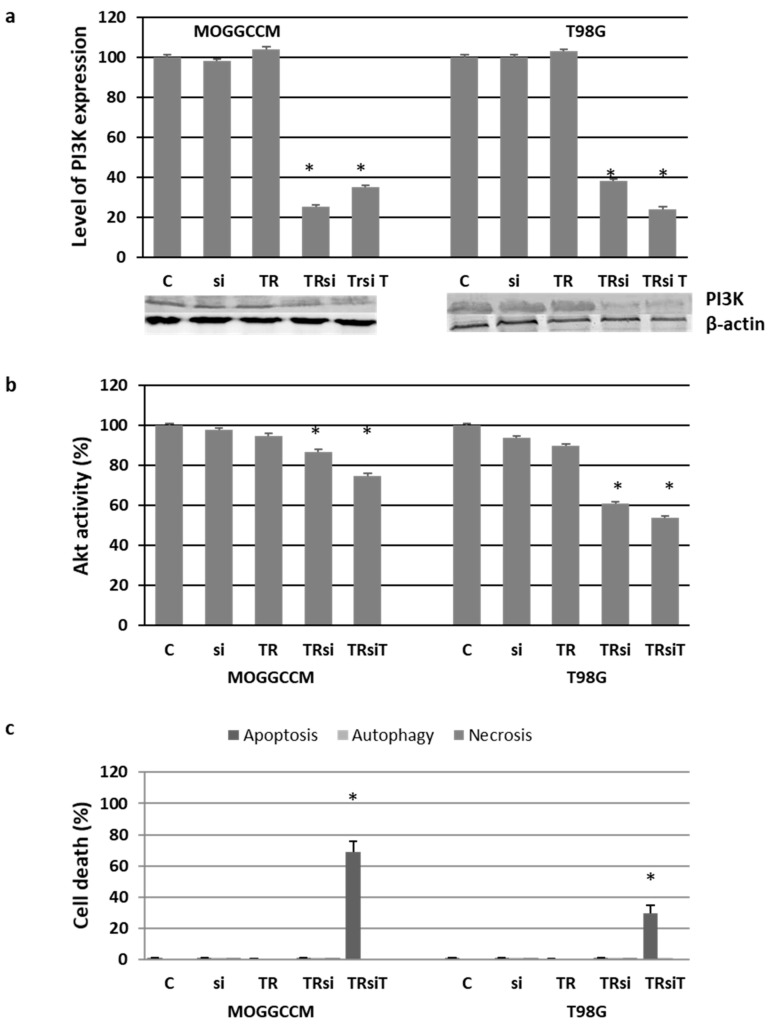
The effect of PI3K blocking by specific siRNA (si) and subsequent temozolomide (T) incubation on the level of PI3K expression with representative blots (**a**), Akt activity (**b**), and cell death induction identified microscopically after staining with Hoechst 33342, acridine orange, and propidium iodide, respectively (**c**), in MOGGCCM and T98G cells. C—control; TR—cells incubated with transfection reagent only; TRsi—cells incubated with transfection reagent and siPI3K; TRsiT—cells incubated with transfection reagent, siPI3K, and temozolomide; * *p* < 0.01 compared to control.

**Figure 8 ijms-22-05155-f008:**
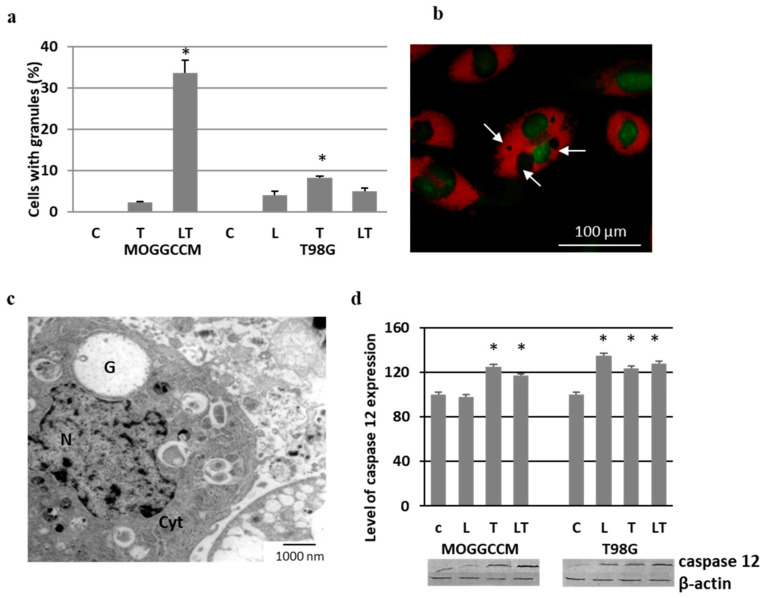
Granule formation in MOGGCCM and T98G cells upon LY294002 (L) and temozolomide (T) treatment. (**a**) Quantitative analysis; (**b**) picture of granules in MOGGCCM cell after LT incubation, not stained with acridine orange and indicated by arrows; (**c**) MOGGCCM electron micrograph with granule (G) after LT treatment (N—nucleus, Cyt—cytoplasm); (**d**) level of caspase 12 expression with representative blots. C—control, * *p* < 0.01 compared to control.

**Figure 9 ijms-22-05155-f009:**
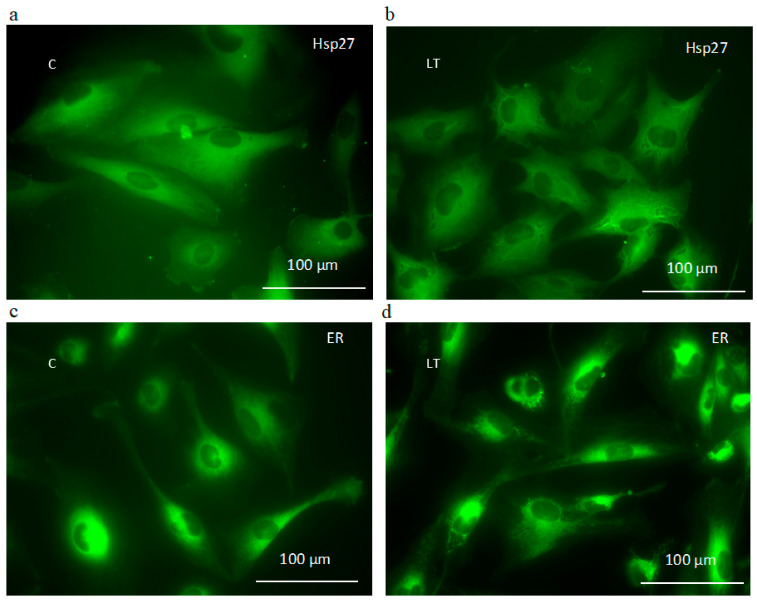
Localization of Hsp27 detected immunocytochemically with specific anti-Hsp27 antibodies (**a**,**b**) and ER structure after staining with DiOC_6_(3) (**c**,**d**) in control (**a**,**c**) and LY294002 (L) with temozolomide (T) treated (**b**,**d**) MOGGCCM cells. C—control.

**Figure 10 ijms-22-05155-f010:**
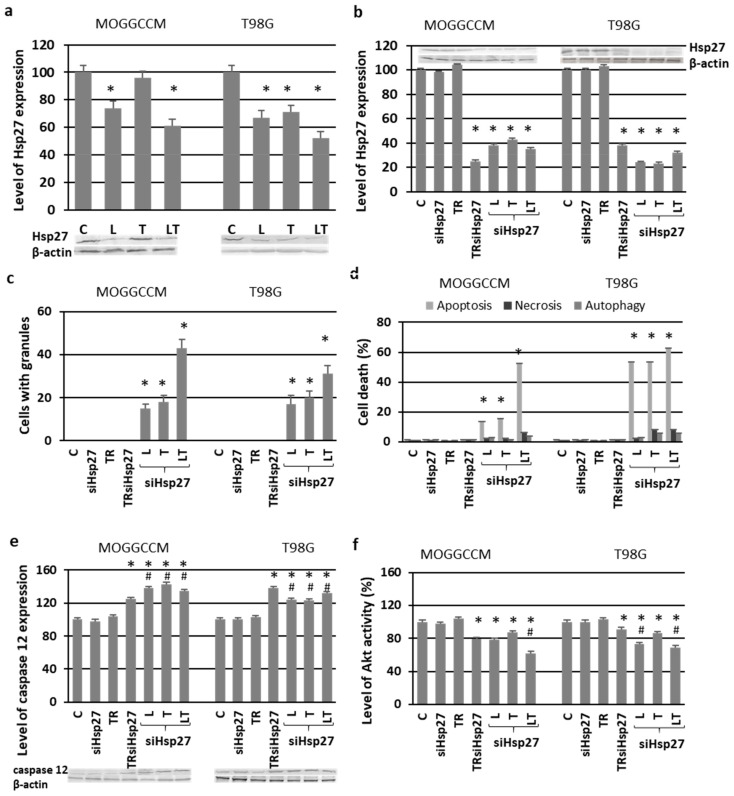
The effect of Hsp27 blocking by specific siRNA (siHsp27) and subsequent temozolomide (T) and LY294002 (L) incubation on the level of Hsp27 expression with representative blots (**a**,**b**); granule formation estimated microscopically (**c**); apoptosis, autophagy, and necrosis identified microscopically after staining with Hoechst 33342, acridine orange and propidium iodide, respectively; (**d**) the level of caspase 12 with blots (**e**); and Akt activity (**f**). C—control, TR—cells incubated with transfection reagent only, TRsiHsp27—cells incubated with transfection reagent and siHsp27, * *p* < 0.01 compared to control, # *p* < 0.01 compared to TRsiHsp27.

## Data Availability

The data presented in this study are available from the corresponding author.

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
