# Peer review of "Involvement of PI3K Pathway in Glioma Cell Resistance to Temozolomide Treatment"

_ijms, 2021, doi:10.3390/ijms22105155_

Round 1

Reviewer 1 Report

The paper tried to investigate the anticancer potential of a PI3K inhibitor drug  LY294002 when used in combination with Temozolomide. Using a series of experiments on a glioblastoma cell line and an anapestic astrocytoma cell lines, they checked the effect of the proposed drug combination on cell apoptosis, autophagy, necrosis and migration. The effect of the drugs was observed on caspase, PI3K, mTOR and Akt expression, Akt activity and cytoplasmic granule formation. The effect of PI3K inhibition and Hsp27 inhibition in combination with the drugs are also investigated. 

While the question pursued here is of importance, the presentation of the results is vague. It cannot be said that the figures presented in the paper clearly support the conclusions. Also, lack of explanations in the figure legends make them hard to follow.

Author Response

Thank you very much fo the review of our article. We would like to inform that several changes have been made throughout the text of the manuscript to improve the quality of presented results and their description. Figures have been explained in more detail. All the changes have been marked in red. Several changes have been made throughout the text of the manuscript to improve the quality of presented results and their description. Figures have been explained in more detail. All the changes have been marked in red. 

Reviewer 2 Report

In the manuscript by Adrian ZajÄ…c et al., the authors investigated the effect of LY294002 (PI3K inhibitor) on the efficacy of Temozolomide therapy in human glioma cells. In the study, the authors focused on apoptosis and migratory potential in Human glioblastoma multiforme cells and human anaplastic astrocytoma cells after the Temozolomide treatment with or without LY294002. The molecular mechanism of drug resistance in glioma cells also provided, nevertheless, there are several limitations regarding the study. Overall, the study is conceptually not well supported and needs novel methodological approaches. Besides, the manuscript in its current form is confusing, thereby making it difficult to interpret and even read at times. Importantly, data presentation and statistics are poorly demonstrated which leads the reader to a critical vision of the overall study. I do not believe this study is of sufficient quality to warrant publication in IJMS in its present format.
Some major suggestions are as follows before considering the manuscript for revision:

1.    All the graphs should be prepared using GraphPad Prism and clearly presented.
2.    I would recommend careful statistical analysis by a statistician.
3.  Western blot images should be properly sized and with analyzed molecular weight.

Author Response

Thank you very much for the review of our article. Additionally we would like to inform that throughout the manuscript several changes have been included to improve the flow of the text and to explain results and methods more precisely for clarity. All the changes are marked with red.

Comment 1.    All the graphs should be prepared using GraphPad Prism and clearly presented.

Answear: We decided not to prepare all graphs in another program but we increased the quality of already presented Figures.

Comment 2.    I would recommend careful statistical analysis by a statistician.

Answer: A one-way Anova test followed by Dunnett’s multiple comparison analysis was used for statistical evaluation. This is a classic analysis commonly used in this type of research, therefore we decided not to change the type of analysis. Besides, the scientific team includes people with the necessary skills to perform such calculations, which guarantees their reliability.

Comment 3.  Western blot images should be properly sized and with analyzed molecular weight.

Answer: Molecular weights of studied proteins were not included in the images on purpose, not to overload the already complex and detailed figures. If, in the opinion of the Reviewer, such information increases the value of the work, we will do so immediately.

Reviewer 3 Report

The authors of the study entitled: “Involvement of PI3K pathway in glioma cells resistance to Te- mozolomide treatment” studied the effect of Temozolomide and an inhibitor of the PI3K pathway on glioma cells. The authors used several techniques to investigate the effect of the studied molecules on cell death, migration, caspases 3, 8, 9, and AKT activities, protein levels of Beclin 1, PI3K, mTOR, AKT, caspase 12 and HSP27. The results obtained are interesting, but some methods and analyzes should be improved. The main comments are as follows:

The flow of the results section should be improved.

The citations in the introduction are at the end of the paragraph. The reader will be helped to have a reference at the appropriate place where it is cited. E.g. (p2/line60) “One of the PI3K inhibitors is LY294002, a classic ATP-competitive agent which acts by binding to ATP-binding cleft of the lipid kinase [ref].”

Information on fluorescent probes or assays used to measure apoptosis, autophagy, and necrosis is missing in the legends of Figures 1 and 2. It will be useful for the reader to know this information while reading the images rather than looking in the material and methods at the end of the manuscript. An illustrative image would be appropriate. If the staining was with Hoechst, propidium iodide and acridine orange, the info should be introduced and demonstrated in Figure 2c,d.

P4/lines 91-103 The abbreviation TMZ is used in this paragraph (and further in text). It was not introduced. Could it be Temozolomide? In particular, it is not clear in the following sentence (line 101): “…seemed that TMZ reduced such an effect (only 2% after Temozolomide…”. Please check this throughout.

I would expect a more detailed description of the results obtained in Figure 3, not just one summary sentence. In addition, LY294002 reduced cell migration more than Temozolomide. There is also a difference between the two cell lines.

Could you mention the methods used to obtain the results when describing them? It will help the reader understand, for example, the estimation of caspase activities and protein level of Beclin 1, because another method was used to obtain them.

Molecular wight is missing from Western Blot images.

P7/line 154 What is a difference between LY294002 and PI3K inhibitor? Should it be the same molecule? Was another inhibitor used in this study? This is not shown in Figure 6d.

Figure 6 demonstrates the level of proteins in the cells, not the expression of PI3K and Akt.

Figure 7c should demonstrate: “apoptosis induction in the transfected cells after the Temozolomide treatment.“ However, the description in the image is probably incorrect. Apoptotic cells are demonstrated in T treated cells.

Could you indicate what kind of granules you are watching in section 2.5? Acridine orange staining is usually used to label autophagic vesicles.

P8/line 188 The abbreviation OA should be AO and should be introduced at the place where acridine orange first appears.

Figure 8: An electron microscope image should be described. The displayed organelles as well as granules should be identified. The red and green colors presented in the fluorescent image should be identified. AO is ratiometric fluorescent probe. Acidic vesicles usually correspond to autophagosomes and should be labelled with AO.  

Figure 9: Images should be stained with a nuclear probe (Hoechst, DAPI…) to distinguish between the nuclei and granular structure described in lines 199-201. The authors reported that the vesicles were surrounded by Hsp27, but in Figure 9a, b this is not visible. Similarly, the fluorescent images of ER structures are so scattered that it is not possible to observe a granular structure near the vesicular structure from these images. ER localization is concentrated in the perinuclear region of the studied cells. 9a and c are identified as control, but marked LT and vice versa b,d.  In general, there is no difference between a and b, and c and d. It is not clear whether live staining or immunostaining was used to visualise the ER in these cells.

Figure 10 : The description of this figure is not clear. What are siHsp27 and TRsiHsp27? The legend identifies si cells and TRsi. That should be accurate. I also propose a correction of -Hsp27 to siHSP27, because it is gene silencing and not knock out. What is the level of significant difference between TRsiHsp27 and L, T and LT treatment in e,f?

Lines 323-326 The first sentence of the conclusion is not supported by the results. No evidence of flavonoid effect was demonstrated.

The fluorescence microscopy specification is missing in the material and methods section. What was the excitation of the samples (fluorescent probes), the emission, the objective used, the camera? It is not clear from the results how you distinguished between the cells and identified them as apoptotic, necrotic and autophagic. I suggest describing it in either results or methods section.

What was the channel and excitation for DiOC6 analysis by flow-cytometry? Parameters are also missing for ER staining. How can you distinguish between mitochondria and ER staining? What was the source for reducing DiOC6 fluorescence in flow-cytometry? Mitochondrial or ER stress? How do we know it was a decrease in mitochondrial potential? You also mentioned that the same treatments caused ER stress. These measurements are confusing.

If the reader is not familiar with May-Grünwald-Giemsa method, the brief principle should be introduced.

P15/408 the samples were not measured calorimetrically, but probably colorimetrically.

P15/417 Was FITC conjugated to primary or secondary antibody?

Protein localization was probably analysed using confocal fluorescence microscope PASCAL5(Zeiss) and not scanning head. What does the wavelength of 488 nm belong to? Excitation laser, emission detection window?

The quality of the images is low and should increased.

Author Response

We would like to thank the Reviewer for a very valuable and detailed opinion of our work. At the same time, we inform that all changes in the text were marked in red.

Comment 1. The citations in the introduction are at the end of the paragraph. The reader will be helped to have a reference at the appropriate place where it is cited. E.g. (p2/line60) “One of the PI3K inhibitors is LY294002, a classic ATP-competitive agent which acts by binding to ATP-binding cleft of the lipid kinase [ref].”

Answer: Appropriate reference has been placed at the end of the sentence “One of the PI3K inhibitors is LY294002, a classic ATP-competitive agent which acts by binding to ATP-binding cleft of the lipid kinase [12,13].

Comment 2. Information on fluorescent probes or assays used to measure apoptosis, autophagy, and necrosis is missing in the legends of Figures 1 and 2. It will be useful for the reader to know this information while reading the images rather than looking in the material and methods at the end of the manuscript. An illustrative image would be appropriate. If the staining was with Hoechst, propidium iodide and acridine orange, the info should be introduced and demonstrated in Figure 2 c,d.

Answer: Appropriate information of the dyes used to dead cells identification has been placed in the legends of Figures 1 and 2, according to the Reviewer suggestion, as well as in the description of Figure 7 and Figure 10. Additionally, in the Materials and Methods section detailed information about typical for apoptosis, autophagy and necrosis morphology observed under the microscope upon adequate staining has been included.

Comment 3. P4/lines 91-103 The abbreviation TMZ is used in this paragraph (and further in text). It was not introduced. Could it be Temozolomide? In particular, it is not clear in the following sentence (line 101): “…seemed that TMZ reduced such an effect (only 2% after Temozolomide…”. Please check this throughout.

Answer: Indeed, the abbreviation of Temozolomide - TMZ used in some parts of the manuscript was confusing. Therefore, for clarity, we decide to use the full name of the drug throughout the text only.

Comment 4. I would expect a more detailed description of the results obtained in Figure 3, not just one summary sentence. In addition, LY294002 reduced cell migration more than Temozolomide. There is also a difference between the two cell lines.

Answer: More detailed description of the effect of LY294002 and Temozolomide on the migratory potential of glioma cells has been included in 2.1 section.

Comment 5. Could you mention the methods used to obtain the results when describing them? It will help the reader understand, for example, the estimation of caspase activities and protein level of Beclin 1, because another method was used to obtain them.

Answer: For clarity, the Results were enriched with the short information about the methods used to obtain presented data.

Comment 6. Molecular wight is missing from Western Blot images.

Answer: Molecular weights of studied proteins were not included in the images on purpose, not to overload the already complex and detailed figures. If, in the opinion of the Reviewer, such information increases the value of the work, we will do so immediately.

Comment 7. P7/line 154 What is a difference between LY294002 and PI3K inhibitor? Should it be the same molecule? Was another inhibitor used in this study? This is not shown in Figure 6d.

Answer: This is the same molecule and indeed in this context may sound confusing. The sentence has been changed into “TMZ alone did not change the level of the protein but, surprisingly, significantly inhibited the activity of Akt. It was also observed after LY294002 treatment, both alone and in combination with Temozolomide (Figure 6d).”

Comment 8. Figure 6 demonstrates the level of proteins in the cells, not the expression of PI3K and Akt.

Answer: Figure 6 description has been changed from “Level of PI3K (a), Akt (b) and mTOR (c) expression and Akt activity (d)….” to “Level of PI3K (a), Akt (b) and mTOR (c) with representative blots as well as Akt activity (d)….”

Comment 9. Figure 7c should demonstrate: “apoptosis induction in the transfected cells after the Temozolomide treatment.“ However, the description in the image is probably incorrect. Apoptotic cells are demonstrated in T treated cells.

Answer: The description of Figure 7c was not correct: it was “T” but should be “TRsiT” and such change has been made in the Figure 7c diagram.

Comment 10. Could you indicate what kind of granules you are watching in section 2.5? Acridine orange staining is usually used to label autophagic vesicles.

Answer: Acridine orange staining of MOGGCCM and T98G cells treated with LY294002 and Temozolomide revealed the presence within the cytoplasm rounded structures that we called “granules”.  But in contrast to autophagic vacuoles, which were stained with AO, “granules” were not stained with this dye and looked as “wholes” within the cytoplasm. We observed such structures for the first time after treatment with Temozolomide and quercetin (Jakubowicz-Gil et al, 2013, Tumor Biology, 34:236-2378) and proper explanation was placed in the Discussion section. But we agree that in the Results section 2.5 information that AO staining revealed their presence may be confusing and therefore information that those granules were not stained with AO has been placed for clearance at the begging of the section. Similarly, description of Figure 8 has been changed.

Comment 11. P8/line 188 The abbreviation OA should be AO and should be introduced at the place where acridine orange first appears.

Answer: Abbreviation OA has been changed onto AO and appeared for the first time with its full name in the Results section 2.1.

Comment 12. Figure 8: An electron microscope image should be described. The displayed organelles as well as granules should be identified. The red and green colors presented in the fluorescent image should be identified. AO is ratiometric fluorescent probe. Acidic vesicles usually correspond to autophagosomes and should be labelled with AO. 

Answer: Electronogram has been labelled with G – granule, N – nucleus, Cyt – cytoplasm and appropriate explanation has been placed in the Figure 8 legend. More precise information that granules presented at Figure 8b were not stained with acridine orange has also been included.

Acridine orange may interact with DNA and then it is spectrally very similar to fluorescein. AO and FITC have a maximum excitation at 502nm and 525 nm (green) and using Nikon E800 with excitation filter G 2A Ex 510-560 it is possible to observed green DNA and red cytoplasmic areas with low pH including autophagic bodies (AVOS). It is also possible to observed granules (not stained with AO). With excitation filter B 2A Ex 450-490 only compartments with low pH can be seen (without DNA) like AVOS and additionally granules. Because  sometimes granules give similar picture with nucleus under the microscope to distinguish those two structures/organelles we decided to use filter G 2A Ex 510-560 in experiments with granules and filter B 2A Ex 450-490 when we needed to observe only AVOS.

Comment 13. Figure 9: Images should be stained with a nuclear probe (Hoechst, DAPI…) to distinguish between the nuclei and granular structure described in lines 199-201. The authors reported that the vesicles were surrounded by Hsp27, but in Figure 9a, b this is not visible. Similarly, the fluorescent images of ER structures are so scattered that it is not possible to observe a granular structure near the vesicular structure from these images. ER localization is concentrated in the perinuclear region of the studied cells. 9a and c are identified as control, but marked LT and vice versa b,d.  In general, there is no difference between a and b, and c and d. It is not clear whether live staining or immunostaining was used to visualise the ER in these cells.

Answer: Figure 9 has been arranged once again for better visualization of the Hsp27 localization and ER structure. 

Comment 14. Figure 10: The description of this figure is not clear. What are siHsp27 and TRsiHsp27? The legend identifies si cells and TRsi. That should be accurate. I also propose a correction of -Hsp27 to siHSP27, because it is gene silencing and not knock out. What is the level of significant difference between TRsiHsp27 and L, T and LT treatment in e,f?

Answer: Figure 10 description has been corrected according to the Reviewer suggestions. Additionally, significant changes comparing TRsiHsp27 and L, T and LT samples in transfected glioma cells has been included and presented as #.

Comment 15. Lines 323-326 The first sentence of the conclusion is not supported by the results. No evidence of flavonoid effect was demonstrated.

Answer: Indeed, the meaning of the sentence was misleading, as the word “derivative” was missing at the end of it. For clarity, the sentence has been redrafted and simplified.

Comment 16. The fluorescence microscopy specification is missing in the material and methods section. What was the excitation of the samples (fluorescent probes), the emission, the objective used, the camera? It is not clear from the results how you distinguished between the cells and identified them as apoptotic, necrotic and autophagic. I suggest describing it in either results or methods section.

Answer: More precise deception of microscopy specifications as well as morphological changes characteristic for apoptosis, necrosis, autophagy and cytoplasmic granules have been included in the Materials and Methods section, “Microscopic detection of apoptosis, autophagy, and necrosis with fluorochromes”

Comment 17. What was the channel and excitation for DiOC6 analysis by flow-cytometry? Parameters are also missing for ER staining. How can you distinguish between mitochondria and ER staining? What was the source for reducing DiOC6 fluorescence in flow-cytometry? Mitochondrial or ER stress? How do we know it was a decrease in mitochondrial potential? You also mentioned that the same treatments caused ER stress. These measurements are confusing.

Answer: DiOC6(3) is a cell-permeant, green-fluorescent, lipophilic dye that is employed to stain mitochondria and endoplasmic reticulum in animal and plant cells. Its selectivity depends on the concentration used. When applied at low concentration it is selective for the mitochondria but at higher concentration may be used to stain the endoplasmic reticulum

- 3,3′-Dihexyloxacarbocyanine iodide at Sigma-Aldrich or ThermoFisher;

- Terasaki M. (1989) Fluorescent labeling of endoplasmic reticulum. Methods Cell Biol. 29: 125–35;

- Terasaki M. et al (1992) Characterization of endoplasmic reticulum by co-localization of BiP and dicarbocyanine dyes. J Cell Sci 101:315-322;

- Koning A. et al (1993) DiOC6 staining reveals organelle structure and dynamics in living yeast cells.  Cell Motil Cytoskeleton 25 (2): 111–28;

- Kim E.J. et al (2008) Underlying mechanism of quercetin-induced cell death in human glioma cells. Neurochem Res 33:971-979;

Additional parameters considering flow cytometry and microscopic analysis have been included in the appropriate Materials and Methods sections.

Comment 18. If the reader is not familiar with May-Grünwald-Giemsa method, the brief principle should be introduced.

Answer: May-Grünwald-Giemsa method has been briefly described at the Materials and Methods section, “Cell migration test” 

Comment 19. P15/408 the samples were not measured calorimetrically, but probably colorimetrically.

Answer: “Calorimetrically” has been changed to “colorimetrically”

Comment 20. P15/417 Was FITC conjugated to primary or secondary antibody?

Answer: Secondary antibodies were FITC conjugated and appropriate information has been placed in the Materials and Methods section, “Indirect immunofluorescence”.

Comment 21. Protein localization was probably analysed using confocal fluorescence microscope PASCAL5(Zeiss) and not scanning head. What does the wavelength of 488 nm belong to? Excitation laser, emission detection window?

Answer: Imprecise two sentences (“Protein localization in the cells was analyzed using the scanning head PASCAL5(Zeiss). The fluorescent channel was λ=488nm.”) were supplemented with the information suggested by the Reviewer and redrafted into one sentence: “Protein localization in the cells was analyzed using confocal fluorescent microscope PASCAL5(Zeiss) with the excitation wavelength for fluorescein λ=488nm.”

Comment 22. The quality of the images is low and should increased.

Answer: The quality of the images has been increased

Round 2

Reviewer 2 Report

The authors have not satisfactorily addressed the comments I raised. Following a careful review of the results presented in the manuscript, I do not find the current format of the manuscript suitable for publication. My primary concern is the representation of the results that leads to a critical view for the overall study.

Author Response

Thank you for evaluating the presented manuscript. We would like to inform
that the figures in the manuscript have been changed for editorial purposes
.
We also decided not to change the structure of the manuscript and leave
the content unchanged.

Reviewer 3 Report

The authors address all of issues raised by me. I am overall satisfied with the revision.

Author Response

Thank you for your review and approval of the changes made.